# A Review of Bioactive Factors in Human Breastmilk: A Focus on Prematurity

**DOI:** 10.3390/nu11061307

**Published:** 2019-06-10

**Authors:** Andrea Gila-Diaz, Silvia M. Arribas, Alba Algara, María A. Martín-Cabrejas, Ángel Luis López de Pablo, Miguel Sáenz de Pipaón, David Ramiro-Cortijo

**Affiliations:** 1Department of Physiology, Faculty of Medicine, Universidad Autónoma de Madrid, 28029 Madrid, Spain; andrea.gila@estudiante.uam.es (A.G.-D.); silvia.arribas@uam.es (S.M.A.); alba.algara@estudiante.uam.es (A.A.); angel.lopezdepablo@uam.es (Á.L.L.d.P.); 2Department of Agricultural Chemistry and Food Science, Universidad Autónoma de Madrid, Institute of Food Science Research, CIAL (UAM-CSIC), 28049 Madrid, Spain; maria.martin@uam.es; 3Neonatology Service, La Paz Hospital-Universidad Autónoma de Madrid, 28046 Madrid, Spain; miguel.saenz@salud.madrid.org; 4Carlos III Health Institute, Maternal and Child Health and Development Research Network, 28029 Madrid, Spain

**Keywords:** adipokines, antioxidants, breastfeeding, cytokines, growth factors

## Abstract

Preterm birth is an increasing worldwide problem. Prematurity is the second most common cause of death in children under 5 years of age. It is associated with a higher risk of several pathologies in the perinatal period and adulthood. Maternal milk, a complex fluid with several bioactive factors, is the best option for the newborn. Its dynamic composition is influenced by diverse factors such as maternal age, lactation period, and health status. The aim of the present review is to summarize the current knowledge regarding some bioactive factors present in breastmilk, namely antioxidants, growth factors, adipokines, and cytokines, paying specific attention to prematurity. The revised literature reveals that the highest levels of these bioactive factors are found in the colostrum and they decrease along the lactation period; bioactive factors are found in higher levels in preterm as compared to full-term milk, they are lacking in formula milk, and decreased in donated milk. However, there are still some gaps and inconclusive data, and further research in this field is needed. Given the fact that many preterm mothers are unable to complete breastfeeding, new information could be important to develop infant supplements that best match preterm human milk.

## 1. Introduction

Preterm infants are those born before 37 weeks of gestation [1]. They are considered extremely preterm at less than 28 weeks, very preterm between 28 and 32 weeks, and moderate to late preterm between 32 and 37 weeks [2].

Prematurity is a serious health problem worldwide, with increasing rates in both low- and high-income countries [2,3]. The World Health Organization (WHO) estimated that the global incidence of preterm delivery was around 11.1% in 2010. The frequency of preterm births (PTBs) in low-income settings is between 12% and 18.1% [2], with Asia and Africa being the areas with the highest rates [4]. The frequency of PTB can vary between 5% and 13% in high-income areas such as the United States of America [5]. 

PTB is a multifactorial syndrome and has multiple etiologies. The main factors that may increase women’s risk are maternal malnutrition, poor pregnancy weight gain, infections and maternity at extreme ages, short interpregnancy intervals, or/and obstetric complications. Infertility and the subsequent need for assisted reproduction techniques resulting in twin pregnancies are also important factors. Finally, exposure to lifestyle-related toxic substances and environmental pollutants are also considered as potential risk factors [3,4,5,6,7,8]. 

Preterm infants, in comparison with term infants, have higher morbimortality rates. Throughout the world, prematurity is the second most common cause of death among children under 5, and is the leading cause in high-income countries [4,9]. Late onset sepsis [10], necrotizing enterocolitis (NEC) [11], retinopathy of prematurity (ROP) [12], bronchopulmonary dysplasia (BPD) [13], and neurodevelopmental problems [14] are among the most frequent morbidities. Besides, growing evidence is proving that individuals born preterm are at higher risk of cardiovascular and metabolic diseases in adulthood [15,16,17].

Nutrition during the earliest stage of life is of great importance for the growth and maturation of tissues and organs, especially for those infants with PTB. Breastmilk (BM) is tailored to cover the needs of the newborns, providing the adequate amount of macro and micronutrients for infant growth and development. In addition, it contains several bioactive compounds, which contribute to the maturation of their immune system, among other important aspects. The BM composition is dynamic and varies according to the maternal age, number of pregnancies [18], body mass index (BMI) [19], maternal diet, time of the day, lactation period, and other environmental factors [20]. Therefore, the WHO recommends that infants be exclusively breastfed for the first six month of life, and lactation, together with nutritionally adequate foods, can continue up to two years of age or beyond [21,22]. This recommendation has been endorsed by several Pediatric and Nutrition Associations [23,24,25]. 

Among the most important bioactive compounds found in BM, the most outstanding are antioxidants, which may help counteracting the negative effects of oxidative stress in newborns [26]. Other bioactive compounds to bear in mind are growth factors and hormones, which regulate the energy intake and maturation of organs. Also, cytokines are present in BM, which may help to protect against infections or reduce inflammatory processes [20]. These bioactive compounds may contribute to the long-term protection of preterm infants against the development of cardiometabolic diseases. Premature infants fed with BM exhibit lower rates of metabolic syndrome, hypertension, or insulin resistance in adolescence, compared with the newborns who are not breastfed [24,27].

BM is the ideal food for neonates during the first days of life and cannot be equaled by artificial substitutes [28]. However, there are mothers who, for several reasons, do not breastfeed their newborns. In 2018, 7.6 million babies were not breastfed [29]. In these cases, commercial formulas and donated BM, generally from mothers with term deliveries [30], are used. The human milk is recognized as a better scavenger of free radicals than the infant formula due to its bioactive compounds, which are lacking in commercial formulas [31,32].

The aim of this review is to summarize the role of some bioactive factors present in BM, namely antioxidants, growth factors, adipokines, and cytokines, with specific attention to the differences between preterm and full-term human milk; we also summarize their role on the development and the potential beneficial actions on neonatal and long-term health of premature infants. 

## 2. Bioactive Compounds in Breastmilk

There is wide evidence that BM prevents many of the perinatal complications associated with preterm labor [33], and BM is recognized as a protective factor against morbidity and mortality. Some of the beneficial factors against preterm complications may be related to the bioactive compounds and interactive elements present in the milk, such as antioxidants, growth factors, adipokines, cytokines, or antimicrobial compounds [34].

### 2.1. Antioxidants in Human Milk

Reactive oxygen species (ROS) are physiologically relevant molecules that participate in cellular signaling processes. However, ROS are highly oxidizing and, in excess, can cause damage to cellular structures. To counteract the oxidative effects of ROS, there are a wide variety of antioxidant systems. There is a delicate balance between ROS and other reactive species and antioxidants. If this balance is lost, the result is oxidative stress. This can be due to an excessive ROS production that exceeds the antioxidant capacity, or insufficient antioxidant systems. 

Birth represents a significant oxidative challenge because of the switch from the relatively low-oxygen intrauterine environment to the high-oxygen extrauterine atmosphere [35]. Thus, newborns are exposed to an increase in ROS during labor and the transition to neonatal life [36]. PTB disrupts the normal developmental upregulation of antioxidant systems. Increased oxidative stress is observed in preterm neonates compared with full-term infants [37], being a critical factor that exacerbates perinatal morbidities of prematurity [38].

Endogenous antioxidants can be classified as enzymatic (i.e., superoxide dismutase (SOD), catalase, or glutathione peroxidase (GPx)), small non-enzymatic molecules (like glutathione (GSH)), or hormones with antioxidant capacity (such as melatonin) [37,39]. In addition to endogenous antioxidants, several foods, mainly vegetables and fruits, contain antioxidants such as vitamins, carotenoids, and polyphenols, among others. 

#### 2.1.1. Antioxidant Properties of Breastmilk 

BM has a powerful antioxidant composition and all the above-mentioned compounds have been found in it [18,19,26,40,41]. Antioxidants are important for newborn protection against disease [28], and may be critical for infants with PTB. There is evidence that premature neonates nourished with BM have less oxidative stress, evidenced by the lower levels of oxidative damage biomarkers compared with the infants who were formula fed [40,42]. It has been proposed that the reduction in oxidative damage by BM may be related to the reduction in ROS synthesis—evidenced for hydroxyl radicals—or to an increase in the antioxidant defense systems [42]. Some reports show that preterm infants fed with BM plus a fortifier had higher antioxidant urinary levels compared to both infants exclusively fed BM and with those who were formula-fed; the mechanisms remain unclear [43] and this aspect should be further analyzed. 

The total antioxidant capacity of BM seems to be higher in colostrum compared to mature milk [44,45] and its radical scavenging activity decreases along the lactation period. Pasteurization processes and storage conditions of BM may reduce antioxidants [46], as well as some immunological and nutritional properties. For example, refrigeration decreases the concentration of vitamins, lactoferrin, and lipases. Therefore, it is not recommended for BM. However, despite the fact that these treatments alter the bioactive compounds, donated milk should be pasteurized and frozen. As stated by the Spanish Association of Pediatrics, the development of methods to improve the preservation of the antioxidant capacity of donated BM, or any factors improving maternal antioxidant status, deserve further investigation [47]. BM treatment is important, particularly in the context of prematurity. Certain viruses, such as cytomegalovirus, may infect the newborn [48] through raw BM transmission [49]. The cytomegalovirus infection is problematic in preterm infants, particularly in those with very low birth weight [48]. 

Differences in antioxidant capacity of term and preterm BM remain controversial. Some data show that preterm BM has more antioxidants and equal resistance to oxidative stress as compared to term human milk [43]. On the other hand, other studies show a lower total antioxidant capacity in preterm BM compared to full-term BM [44,50]. It has been suggested that this is due to the fact that antioxidants tend to accumulate over the last three months of gestation. Thus, it is proposed that a mother with a PTB would synthetize a less antioxidants [50,51]. Discrepancies in the antioxidant activity between term and preterm BM may be related to differences in the ethnic group studied or the maternal nutritional habits. Another possible explanation to this controversy could be the time point when the studies were conducted. In this sense, it has been observed that the total antioxidant capacity of premature BM does not decline after one week, and preterm neonates benefit more from the antioxidant capacity of colostrum and transitional milk [50]. Term milk may differ in this respect. Taken together, these reports suggest that more research is needed in this area. Aspects such as ethnicity, maternal nutritional status, or breastfeeding period should be taken into account in future studies to clarify differences in antioxidant properties in BM and its relationship with infant needs.

#### 2.1.2. Antioxidants Present in Breastmilk 

The antioxidant properties of human BM are related to the combination of different compounds, both exogenous and endogenous molecules. 

Several food-derived antioxidants, including polyphenols, carotenoids, and vitamins, have been reported in BM, and there is evidence that they are more abundant in preterm than in full-term BM. The levels of these bioactive components also differ in formula milk. As stated in Table 1, the levels of antioxidants included in formula milk do not always match the reports in BM, and in some instances are clearly much higher. An excess in antioxidants may be deleterious, since some antioxidants may turn into pro-oxidant molecules under certain circumstances. Therefore, research on the adequate requirements of antioxidants in both term and preterm infants and adjustment of the antioxidant content in formula are needed.

Enzymatic antioxidants are also important to counteract oxidative processes. The main enzymatic systems which detoxify ROS are SOD, catalase, and GPx. SOD catalyzes the dismutation of superoxide anion to hydrogen peroxide, which must be removed by catalase. Three SOD isoforms (copper–zinc, manganese, and extracellular SOD) are present in mammals, with different subcellular locations and tissue distribution [52]. Catalase consists of four protein subunits, which make it very resistant to pH changes, thermal denaturation and proteolysis [53]. This enzyme eliminates the hydrogen peroxide generated by SOD and completes the reaction to eliminate ROS [52]. Finally, GPx participates, together with catalase, in the detoxification of hydrogen peroxide among other organic hydroperoxides [54,55].

There is evidence of some of these enzymatic systems and their concentrations have been reported. Catalase has been found in human BM, with some differences between term and preterm BM [19]. In milk from term mothers, the catalase concentration ranges between 0.43 and 0.84 U/mg protein, and in preterm BM the concentration has been found to be 0.5–0.97 U/mg protein [40]. GPx has also been found in BM, with a maximum value of 31.2 mM/min/L [56]. 

Regarding non-enzymatic systems, GSH is a three amino acid peptide, which is the main intracellular low molecular weight antioxidant. It participates in the regeneration of other antioxidants, such as vitamin C and E to their active forms [57,58]. GSH has been found in BM in concentrations ranging from 10.4 to 43.1 nmol/mg. 

Melatonin is the major endocrine product of the pineal gland, which plays a physiological role in neuroimmunomodulation. It is synthetized from tryptophan via serotonin in pinealocytes and many other cells, with a circadian regulation [59,60,61,62,63,64]. Melatonin is interesting in the context of BM bioactive compounds due to its pleiotropic actions. It has been demonstrated to exhibit protective effects against cellular aging due to its antioxidant effects, both as a direct scavenger, and stimulating the expression of SOD, catalase, and GPx [60,65,66,67,68]. On the other hand, melatonin appears to be one of the most promising molecules for neuroprotection in preterm infants due to its effects on the modulation of neuroinflammatory pathways [69]. 

Melatonin is present in BM in much higher concentrations during night time, being almost undetectable during the day. So far, the studies have not been able to detect differences between preterm and term BM mothers. Some studies have observed higher levels in colostrum, which has been reported to increase the phagocytic activity of cells against bacteria. A longer sleep time was observed in newborns who were breastfed than in those who were formula-fed [70]. Table 2 describes several endogenous antioxidants and their concentrations in BM.

### 2.2. Growth Factors in Human Milk

Preterm infants are immature neonates who usually exhibit growth retardation, poor development and physical and neurological deficits. Postnatal growth retardation is likely the result of inadequate nutrition support after delivery, also contributing to poor neonatal health. In this context, in addition to a macronutrient supply, growth factors provided by breastfeeding may be of great importance. Growth factors play a role in the growth, maturation, and integrity of several organs, particularly for the neonatal gastrointestinal tract [71]. They help with the maturation of gut immunity and have anti-inflammatory effects [72,73]. Hirai et al. described the trophic effects of growth factors on fetal and neonatal gastrointestinal tract by promoting the proliferation and differentiation on their immature cells [74]. The highest concentrations of growth factors are provided by colostrum, as the first milk released after birth [75]. 

The main growth factors present in BM and their trophic effects on neonatal organs and systems are summarized in Figure 1.

#### 2.2.1. Hepatocyte Growth Factor (HGF)

HGF was first identified as a potent mitogen of primary cultured hepatocytes. The essential role is promoting organogenesis. It is also involved in the formation of the kidney, lung, mammary gland, teeth, muscle, and neuronal tissues [76]. To preserve proliferation, angiogenesis, and intestinal tissue development through paracrine and endocrine signaling, high levels of HGF are required in BM. This factor is released into BM by multipotent mesenchymal stem cells [77]. In addition to direct proliferative properties, HGF may also regulate the vascular endothelial growth factor (VEGF) synthesis [78]. 

#### 2.2.2. Epidermal Growth Factor (EGF)

EGF is recognized as a critical trophic factor for the normal intestinal cell development (Table 3). The members of the EGF family are first synthetized as transmembrane precursors, eventually undergoing proteolysis into the mature, secreted form of the growth factor [71]. 

Both the amniotic fluid and the BM contain EGF [79,80]. A member of the EGF family is the heparin-binding growth factor (HB-EGF). Its exogenous administration protects from intestinal ischemia-reperfusion injury, hemorrhagic shock, and NEC by enhancing the healing of intestinal anastomoses and reducing anastomotic complications [81,82]. In BM, EGF levels are higher at the beginning of the lactation period and decrease over time. Furthermore, preterm BM contains higher levels of EGF than term BM, which may help in the reduction of NEC incidence [83]. However, very preterm BM contains lower levels than preterm milk, although still higher compared to term milk. In preterm colostrum EGF content has been found in the range of 22.8–373 µg/L and in term colostrum between 27.7 and 209 µg/L [84].

#### 2.2.3. Neuronal Growth Factors

Brain-derived neurotrophic factor (BDNF) is a small neurotrophic protein, which is widely expressed in the mammalian adult brain [85,86]. BDNF, together with S100B protein and glial cell line-derived neurotrophic factor (GDNF), plays a critical role in the development and maintenance of the nervous system, and in neuronal survival and proliferation [87]. BDNF, S100B, and GDNF are present in human milk. S100B protein and GDNF levels increase within the lactation period [86]. 

#### 2.2.4. Insulin-Like Growth Factor (IGF) Superfamily

Human BM contains IGFs such as IGF-I and IGF-II [88]. IGF-I synthesis is regulated by the availability of amino acids and the overall energy intake, and is a marker of the nutritional status. IGF-I levels in BM are higher during the first days after delivery, decreasing as the milk matures [89]. No significant differences were found between preterm and term milk in other growth factors from this family, except for IGF-I and IGF-II, which is higher in preterm milk [88,90]. IGF-I could be important in the protection of enterocytes after intestinal damage caused by ROS [91]. Enteral IGF-I administration enhances erythropoiesis and augments hematocrit, but its function is still not clearly known [20]. 

#### 2.2.5. Vascular Endothelial Growth Factor (VEGF)

VEGF mediates vascularization, which is also controlled by IGF-I. In preterm infants, the relative hyperoxia found in the extrauterine environment inhibits the expression of VEGF, interrupting the growth of retinal blood vessels. Pulmonary immaturity and the subsequent need for oxygen therapy contribute to the susceptibility of the retinal tissue to the oxidative injury and the subsequent development of ROP [92]. 

VEGF levels in BM are higher at the beginning of the lactation period, which helps reducing ROP’s burden during the first days of life. There are some controversies regarding the differences in this growth factor between preterm and term BM. While no differences have been reported in some studies [93], other authors demonstrate both lower and higher levels of VEGF in preterm compared to full-term milk (Table 3) [73,90]. 

#### 2.2.6. CD14 Protein 

CD14 acts as a co-receptor for the detection of bacterial lipopolysaccharide (LPS) [94]. CD14 is a protein with two forms: one is a soluble form (sCD14) and the other is anchored to the cellular membrane (mCD14). This latter membrane-bound form is primarily expressed on the surface of monocytes, macrophages and neutrophils [95,96]. CD14 may have a major implication due to the protection provided against subsequent allergy manifestations [97]. There is some evidence that low levels of sCD14 in BM are associated with eczema development [98]. In addition, other factors such as the newborn genotype and the interaction with the bioactive factor present in BM may also participate in the development of allergies [96].

### 2.3. Adipokines in Human Milk

In addition to growth factors, adipokines constitute another group of compounds present in BM which is important for metabolism and infant growth. These cytokines derived from adipocytes have been demonstrated to modify weight gain and fat and lean body mass in infants in the early postpartum period [100] and have long-term effects on metabolic programming. They are also involved in the regulation of food intake and energy balance [101]. Several adipokines, with opposing actions on food intake and energy expenditure, have been found in BM (summarized in Table 4). Thus, it can be hypothesized that the programming of food intake and body composition may be influenced by the relative concentration of these compounds in BM. Likewise, BM adipokines may also modulate the development of metabolic diseases in adulthood such as obesity, type 2 diabetes mellitus, or insulin resistance [102].

#### 2.3.1. Leptin

Leptin is an anorexigenic hormone encoded in the *ob gene* and mostly synthesized by white adipose tissue, which acts through the arcuate nucleus of the hypothalamus. Leptin minimizes the energy intake and increases the energy expenditure [101], and it plays a role in fetal and neonatal growth [103,104].

Several studies have reported the presence of leptin in BM, which may be produced by various cell types in the mammary tissue. The *ob gene* is expressed in the epithelium of the mammary gland of lactating women and it produces leptin [105]. Leptin is also transported from maternal circulation to BM [106] and it has been reported that plasma and BM levels are directly associated with the fat content and the body mass index [100,107]

Leptin levels in human milk vary over the lactation period [108]. Differences in BM leptin concentrations are also reported comparing whole and skimmed milk. In mothers with term infants, leptin concentration in whole milk ranged between 0.2 to 10.1 ng/mL [109,110], being lower in skimmed milk at between 0.1 to 3.4 ng/mL [111,112]. Leptin levels seem to be higher in BM from mothers with preterm newborns, with concentrations ranging between 0.6 and 5.3 ng/mL in skimmed milk [108,113]. 

Several reports indicate that leptin levels in BM can play an important role in infant growth, and its production by human mammary epithelial cells might be regulated physiologically according to the necessity and the state of the infant [114]. Maternal milk of small, appropriate, and large gestational age infants (SGA, AGA, and LGA) has different leptin levels, especially during the first month of life. Human milk leptin levels are significantly reduced in the SGA neonates compared to AGA and LGA infants, together with a rapid growth during the first postnatal 15 days. It has also been reported that leptin levels in milk of infants with accelerated postnatal growth were lower than in infants with normal growth [115]. These findings suggest that the presence of leptin in BM might play an important role in growth, appetite and regulation of nutrition in infancy, especially during the early lactation period.

#### 2.3.2. Adiponectin

Adiponectin is an orexigenic hormone which regulates lipid and glucose metabolism. Adiponectin enhances insulin sensitivity and stimulates fatty acid oxidation through the activation of AMP-activated protein kinase (AMPK) in peripheral tissues, and inhibits hepatic glucose production [101]. In addition, adiponectin has powerful anti-inflammatory effects, influencing the vascular endothelium. Adiponectin stimulates the food intake through the hypothalamus and reduces the energy expenditure through its central activity [116]. The production of adiponectin is regulated by the peroxisomal proliferator-activated receptor-γ (PPAR-γ), a nuclear receptor expressed in the liver and muscle, which protects against obesity-related insulin resistance [117].

Adiponectin is present in BM, in a range between 4.2 and 87.9 ng/mL [118], being more than 40 times greater than the concentrations of leptin [119]. In the colostrum higher levels have been reported, with concentrations in the range between 2.9 and 317 ng/mL [118], in accordance with studies showing that BM adiponectin levels are negatively associated with the lactation period [120]. Other factors that may influence BM adiponectin levels are the maternal nutritional status and body composition, although this aspect is still controversial. Some studies report a positive association between adiponectin and maternal body fat mass [120]; others demonstrate that the adiponectin concentration in colostrum is markedly dependent on maternal diet and nutritional status during pregnancy, and there are also reports that failed to observe an association with maternal BMI or infant birth weight [121]. The maternal hormonal and inflammatory profile may also alter the BM adiponectin levels [122]. 

BM adiponectin also seems to have an influence on infant growth. In term infants who were breastfed for at least six months, high levels of adiponectin are associated with overweight [123]. These data support that BM adiponectin in the first stages of life could have an important implication in the regulation of infant growth [124], which deserves further consideration.

#### 2.3.3. Resistin

Resistin is an adipocyte-derived hormone, which regulates glucose homeostasis and counteracts the action of insulin in peripheral tissues, inhibits adipocyte differentiation and may function as a regulator of adipogenesis [101]. Resistin has been identified in BM in a range between 0.2 and 1.8 ng/mL [125] and its levels decrease along the lactation period [126]. In the perinatal period, it seems that resistin is not directly involved in the regulation of insulin sensitivity or adipogenesis [127]. It has been suggested that resistin could have a role in controlling fetal growth together with other BM hormones, and could be involved in the appetite regulation and in the metabolic development of infants [104]. Resistin could also have an important role in the regulation of the energy metabolism and adiposity in utero. Higher serum resistin levels have been found in term infants compared to preterm infants [102], and it has been suggested that newborns could benefit from these higher concentrations of circulating resistin, facilitating the production of hepatic glucose and preventing hypoglycemia after delivery [128]. 

#### 2.3.4. Ghrelin

Ghrelin is an amino acid synthesized in several organs from the digestive and nervous system, heart and lungs, the stomach being the main site of production [129]. It is one of the most important orexigenic peptides. However, in addition to the functions related to the regulation of food intake and metabolism, ghrelin has other physiological actions, such as in gastric motility and acid secretion, reduction of insulin secretion, adipogenesis, and cardiovascular function, as well as anti-inflammatory effects. Ghrelin secretion declines in situations of positive energy balance, such as after food intake or in obesity, while increasing when fasting or during weight loss. 

The presence of ghrelin in BM may be a strong factor influencing the feeding behavior, and body composition later in life, through its effects on short-term food intake and long-term body weight [104]. Ghrelin is found in both term and preterm human BM [130], with concentrations in the range of 73–6000 pg/mL [131,132]. A gradual and parallel increase has been observed between post-partum ghrelin plasma levels and the concentration in BM [130]. However, this physiological increase is impaired in women with gestational diabetes and pre-gestational diabetes mellitus [133]. These alterations may alter the infant’s feeding behavior. 

A positive correlation between ghrelin concentrations in mature BM and infant weight gain has been found, suggesting that this hormone is involved in postnatal growth [134]. Savino et al observed that infants who were fed with formula milk had higher serum ghrelin levels than those who were breastfed. They propose that formula-fed infants have a better feeding stimulus, which makes them eat more, and consequently, boosts their growth and weight gain. This can explain the protective effect of BM against the development of obesity in childhood and adulthood [135].

#### 2.3.5. Obestatin

Obestatin drifts from ghrelin and its synthesis is mainly produced by digestive system cells, especially from the stomach and small intestine [136]. Obestatin is an anorexigenic hormone which reduces food intake, regulates weight gain and gastric emptying by suppressing the intestinal motility [125]. It was proposed that, in addition to its effects on energy balance regulation, it opposes the actions of ghrelin [130]. Some data have identified higher levels of obestatin in BM than in maternal blood [130]. The reported range of obestatin in BM is from 0.4 to 1.3 ng/mL [125]. Although it is not completely confirmed, some authors have suggested that colostrum have higher levels of obestatin to prepare the digestive system to receive milk by reducing newborn appetite. [130]. 

#### 2.3.6. Nesfatin

Nesfatin is an anorexigenic [93] neuropeptide related to the melanocortin signaling pathway in the hypothalamus. It is mainly manifested in nervous system cells and peripheral tissues. Nesfatin acts as an appetite regulator and a body fat producer. [137]. Nesfatin has been found in BM in a range between 8 and 14 pg/mL [138].

#### 2.3.7. Apelin

Apelin, an endogenous ligand for the G-protein-receptors [139], participates by maintaining the cardiovascular and fluid homeostasis, regulates appetite, cell proliferation and angiogenesis [140]. Although apelin has been found to regulate food intake, most of the studies have been made on rats, and the results are controversial. More studies in humans are needed to clarify its effect on food intake [140,141,142,143]. Apelin concentrations in BM have been ranged between 43 and 81 pg/mL, being lower in women who developed gestational diabetes [133].

### 2.4. Cytokines in Human Milk

Cytokines are small proteins synthesized by nearly all the nucleated cells [144]. They are signaling molecules involved in the communication between cells [82]. BM is a cytokine-rich food [145]. According to their role in the inflammatory response, cytokines can be divided into those that promote inflammation or protects against infection, and those that decrease inflammation.

A possible role of some cytokines present in BM in preterm infants may be to compensate the delay of the immune system development [146]. It has been hypothesized that immune mediators in human milk may have a powerful role to play in maturating the infant intestine and stimulating the immune system [97]. The stimulation of immune activity by BM cytokines may be related to their capacity to make connections with cells by crossing the intestinal barrier. 

There is large variability between cytokine concentrations among breastfeeding women, being pro-inflammatory cytokines generally low [147]. BM cytokine content can be affected by different factors, such as gestational age, maternal diet, infections, smoking, maternal ethnicity, or exercise [147]. In addition, the cytokine profile of BM fluctuates along the different phases of breastfeeding and the clinical significance of cytokine concentrations in neonatal health outcomes is still under debate [148,149]. 

#### 2.4.1. Anti-Inflammatory Cytokines in Human Breastmilk

The anti-inflammatory cytokines found in BM include transforming growth factor-β (TGF-β), interleukin 7 (IL-7), and IL-10 (Figure 2). IL-7 BM is known to cross the intestinal barrier and contribute to thymus and T-lymphocyte development [150]. In colostrum, IL-10 ranges from 5.9 to 7.3 ng/L in term and from 1.1 to 8.8 ng/L in preterm infants. It has also been reported that these concentrations decrease over lactation, with an average of 0.7 to 1.3 ng/L and 0.5 to 0.9 ng/L for term or preterm mature milk, respectively [81]. The possible differences in IL-10 concentrations between preterm and term milk are still controversial. While some authors have reported lower levels in preterm milk [151], others fail to demonstrate significant differences [152]. An interesting aspect is the finding of a lower IL-10 in BM of infants with increased risk of NEC [153]. These results suggest that the accessibility of this cytokine from BM may be important to modulate the neonatal inflammatory response; however, this is an aspect which deserves further research. 

##### Transforming Growth Factor- β (TGF- β)

TGF-β is an anti-inflammatory cytokine mainly produced by parenchymal cells and infiltrating cells such as lymphocytes, macrophages and platelets [154]. It is found in human milk, with higher levels in colostrum. During breastfeeding, all three isoforms of the TGF-β superfamily are produced, the majority being TGF-β2 [75]. The levels of this cytokine range between 0.1 and 13.3 µg/L in term colostrum and between 1.4 and 43 µg/L in preterm colostrum. These levels decrease along the lactation period, with concentrations in the order of 0.4–2.8 µg/L in term and 0.9–6.3 µg/L in preterm mature milk [84]. 

The TGF-β found in BM is of great importance for the newborn as it suppresses the neonatal T-lymphocytes activity [155], enabling oral and intestinal tolerance. It could also reduce atopic sensitization by controlling the inflammatory processes involved [156]. TGF-β regulates the production of secretory immunoglobulin A (IgA). IgA in human milk confers passive immune protection to the infant [157]. IgA levels are higher in preterm colostrum (1.8–16.4 g/L) than in term colostrum (1.2–11.6 g/L). Levels greatly decrease along the lactation period to an average of 0.2–0.8 g/L [84]. Therefore, TGF-β-mediated tolerance and IgA antibody synthesis play an essential role in the immunological development of newborns. Some studies have shown how maternal exposure to highly microbial environments increased their TGF-β breastmilk concentrations [147], and it has been proposed that the maternal immune mediators are transferred to the infant by the BM. Thus, the increased TGF-β and IgA levels in the very first milk reflect their purpose of targeting microbial antigens and improve the mucosal barrier function [158]. It has been proposed that, during the gestational and lactation period, there is an increased synthesis of TGF-β, due to the infiltration of immune cells into the breast tissue [159]. These studies propose that the local mammary gland production of TGF-β may be responsible for the elevated concentrations observed in human milk.

#### 2.4.2. Inflammatory Cytokines in Human Breastmilk

Most of the inflammatory cytokines, such as tumor necrosis factor alpha (TNF-α), IL-1β, IL-6, IL-8, and interferon gamma (IFNγ), are present at lower concentrations compared to anti-inflammatory cytokines (Figure 3). The levels of these cytokines decrease over lactation [20] and are associated with the timing of delivery [160], being higher in preterm milk than in term BM [82].

IL-6 levels range between 4.4 and 340 ng/L in term colostrum, between 15.3 and 362 ng/L in preterm colostrum, and between 9.3 and 67.9 ng/L in very preterm colostrum. 

IL-8 concentrations vary between 0.04 and 26.3 µg/L in term colostrum, between 0.13 and 14.7 µg/L in preterm colostrum, and between 0.1 and 3.0 µg/L in very preterm colostrum. 

TNF-α levels decrease from an average of 11.4 ng/L in term colostrum to 1.6 ng/L in term mature milk, from 18.2 ng/L in preterm colostrum to 3.2 ng/L in preterm mature milk, and from 4.2 ng/L in very preterm colostrum to 1.8 ng/L in very preterm mature milk [84]. 

There is a controversy about the effect of these cytokines on the health of offspring [149]. Some reports demonstrate that preterm gestation does not substantially influence the cytokine content of BM during the first month of lactation compared to full-term gestation, which can be beneficial for the regional and systemic immune response of the very preterm infant [152]. 

IL-1β is a mediator of the inflammatory response, and is involved in cell proliferation, differentiation, and apoptosis. Several studies have obtained a range of between 2 and 2500 pg/mL for IL-1β in human milk [161]. It has been reported that the higher the levels of IL-1β in human milk, the greater the protection against eczema for infants [96]. 

IFNγ improves the Th1/inflammation response while abolishing the Th2/allergic reaction [145]. The colostrum of allergic mothers contains less IFNγ but more Th2, IL-4 and IL-13 when compared to non-allergic mothers [162].

Other studies have demonstrated that IL-8 and TNFα levels are slightly elevated in advanced BM of mothers who suffered from preeclampsia [163]. However, not all studies reveal a negative effect of the inflammatory cytokines. IL-8 also protects intestinal cells against chemical injury, since it has a trophic function in the developing human intestine [151]. Other authors have shown the relevance of cytokines as risk factors for immunological disease in infants. Thus, the presence of IL-5 and IL-13 in human milk, although extremely low, are risk factors for asthma at the age of one [164]. 

More research is needed on the role of cytokines in BM and how they influence neonatal health. The contribution of cytokines, anti and pro-inflammatory in BM, is different, and this variability could make the difference between health and disease in preterm infants.

## 3. Human Milk Cells

### 3.1. Stem Cells in Human Milk 

Recent data suggest that up to 6% of the cells in human milk are stem cells, and mesenchymal stem cells isolated from BM are potentially reprogrammable to multiple tissue types [165,166]. These cells may be involved in the development of immune cells including the regulatory T cell, which may produce tolerance to non-inherited maternal antigens and suppress anti-maternal immunity. It induces pregnancy microchimerism, leading to intestinal tissue repair and immune development and protection against infectious diseases [166]. 

### 3.2. Leukocytes in Human Milk

Leukocytes are highly present in colostrum, which means that breastfed infants are exposed to up to 1010 maternal leukocytes/day [167]; however the contribution of this exposure in infants’ immune development is not yet clear [148]. A study carried out with 61 mothers and neonates showed a significantly smaller proportion of macrophages in BM of mothers with infants that developed an allergy to cow’s milk. In contrast, a higher content of neutrophils, eosinophils, or lymphocytes was associated with lower allergies to cow’s milk [168]. 

The role of these cells is still far from understood; for example, how these cells pass the stomach and intestinal barriers and access the infant, or their mechanisms of action, remain unclear. Therefore, further research to clarify these aspects of the inflammation and immunity development is required. 

## 4. Human Milk Microbiota

Early microbial colonization is essential for the infant’s metabolic and immunological maturation. Its development begins at birth, and the most important changes occur during the first year of life [169,170]. The microbiome is constantly changing, and it is influenced by hormones, cytokines, and chemokines. After birth, the transference continues along breastfeeding, and it is considered the main cause of variability between exclusively breastfed and formula-fed infants during the first months of life [165,171]. Raw BM is not a sterile food [167] and several reports confirm more than 100 types of viable bacteria/mL in human milk [171], including 65% of the phyla Proteobacteria and 34% of the phyla Firmicutes [82]. Regarding the genera, the most common are *Staphylococcus, Streptococcus, Lactobacillus, Enterococcus, Lactococcus, Weissella, Veillonella*, and *Bifidobacterium* [172,173]. It has been described that human milk microbiota at an early age is strongly linked with other perinatal factors such as place of residence, delivery mode, or maternal food intake [96]. With respect to the comparison between preterm and term milk, few studies have reported a difference in the BM microbiome. Some trends include more *Bifidobacterium* in term milk and more *Enterococcus* in preterm milk [97,167].

The association between the microbiome and various disorders, such as visceral pain, autism spectrum disorder, cardiovascular risk, obesity, depression, or multiple sclerosis, has been well demonstrated. Besides, the microbiota exerts immune-modulating effects that influence allergic reactions [169]. Thus, the gut microbiome from allergic children differs from non-allergic ones in composition and diversity [174]. It has been hypothesized that early gut colonizers could help in developing and maturing the immune system in infants [175]. Probiotic intake throughout gestation and lactation also leads to specific changes in the neonate [176]. A clinical trial reports the beneficial effect of oral supplementation with *Lactobacillus rhamnosus* in women during pregnancy and breastfeeding to reduce the allergy risk in infants. Modulatory effects were observed both on milk composition and the infant gut [177]. Variations in microbiota of preterm infants have been associated with a higher predisposition for developing NEC [178]. Taking together the information available, BM could be considered a probiotic food for infants. However, the potential protective effect of the BM microbiome is not fully understood, and additional research in this area is necessary to understand its role in the newborn’s health.

## 5. Conclusions and Proposal for Future Research

Breastmilk is a complex food and is the gold standard for infant nutrition. The aim of the present review was to summarize current knowledge regarding some important bioactive molecules which participate in growth, defense against oxidative damage, and infections, and are particularly important in prematurity, which is a global rising problem. 

Breastmilk is a dynamic fluid; it is known that its macronutrient composition changes along lactation to match infant needs. The present review provides evidence that bioactive factors also change over time and, in general, the highest levels are found in colostrum. Another relevant conclusion is that bioactive factors are higher in preterm compared to full-term milk. However, the review of the literature evidences some controversy. Regarding antioxidants, there is no consensus in relation to the content and the requirements in preterm and full-term infants, and there is insufficient information on the changes in donated milk due to thermal treatments. The growth factors and adipokines present in breastmilk, together with macronutrients, contribute to organ maturation and development. The issues that deserve further investigation include how maternal dietary habits or metabolic status influence the content of these bioactive compounds. Accessibility of cytokines from breastmilk is important to modulate the immediate and long-term inflammatory response of the infant. There is no consensus on the concentrations of different cytokines in breastmilk and their relative role for neonatal and adult health, an aspect that deserves further investigation. Finally, the immune-modulating effects of breastmilk leucocytes, stem cells, and the microbiota are still far from being understood, and represent an expanding field of research. 

Another conclusion is the insufficient clinical evidence to demonstrate that raw breastmilk is better than donated milk for infant growth. However, due to the better content of bioactive factors, it is possible that it may improve other clinical outcomes such neurodevelopment or prevention of adult diseases. 

Among aspects which require further attention, we suggest several research lines. Firstly, it is necessary to improve and systematize methods to quantify bioactive factors in breastmilk. It would also be of interest to gain insight on how body fat, nutritional status, and health of the mother influence bioactive breastmilk factors. The results obtained from these studies may help to improve clinical practice, particularly in preterm infants, in the progress of treatments to preserve bioactive factors in donated milk, and to design formula milk that best matches infant needs. Increasing knowledge in this area may help to regulate and adjust neonatal infant growth and improve neonatal and long-term health.

## Figures and Tables

**Figure 1 nutrients-11-01307-f001:**
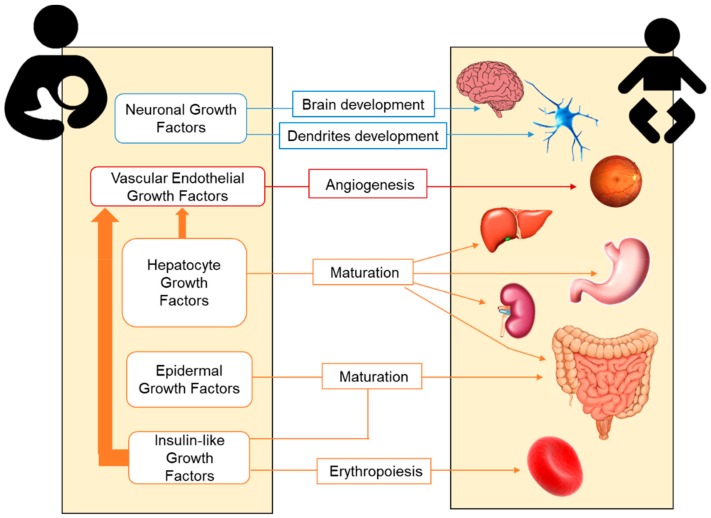
Transference of maternal growth factors through breastmilk and their trophic effects on the growth and maturation of neonatal organs and systems.

**Figure 2 nutrients-11-01307-f002:**
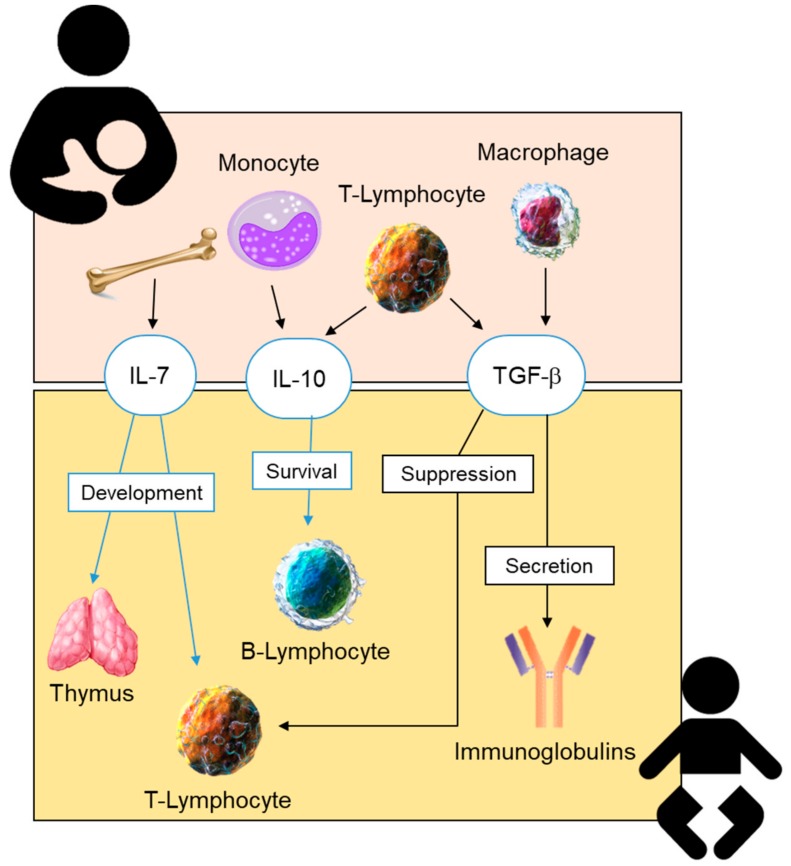
Site of synthesis of anti-inflammatory cytokines present in human breastmilk, and their effects on the neonatal immune system. IL-6, interleukin-6; IL-10, interleukin-10; TGF-β, transforming growth factor-β.

**Figure 3 nutrients-11-01307-f003:**
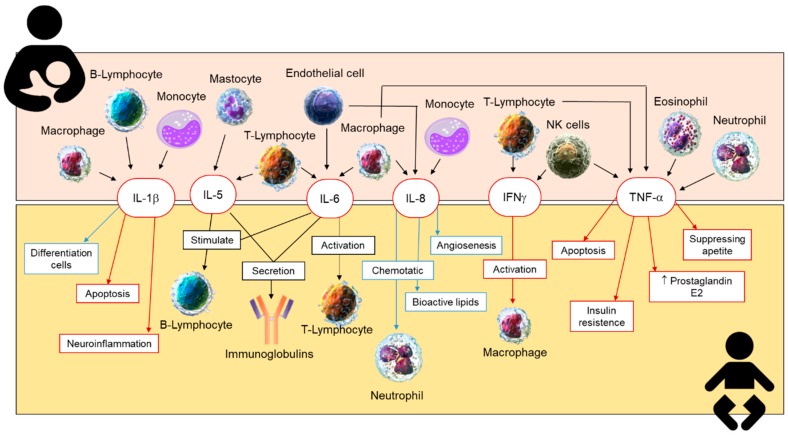
Synthesis of the inflammatory cytokines present in human breastmilk and their effect on the neonate. IL-1β, interleukin-1β; IL-5, interleukin-5; IL-6, interleukin-6; IL-8, interleukin-8; IFNγ, interferon gamma; TNF-α, tumor necrosis factor alpha.

**Table 1 nutrients-11-01307-t001:** Exogenous antioxidants in breastmilk.

Antioxidant Compounds	Preterm Infants	Term Infants	Formula Feeding	
			Preterm	Term
α-carotene	7.7	3.6	0.51	1.40
β-carotene	49.1	13.7	71.1	63.9
Lycopene	66.1	11.9	1.5	5.8
Retinol	401.6	185.8	3086.2	911.8
α-tocopherol	5880.8	1381.9	20,109.1	13,360.2
γ-tocopherol	1207.1	622.8	6787.1	6561.6

Adapted from Hanson et al. (2016) [31]. The data are expressed as μg/L.

**Table 2 nutrients-11-01307-t002:** Endogenous antioxidants in breastmilk.

Antioxidant	Activity	Range
Superoxide dismutase	Eliminates superoxide anion	2.01–6.26 nmol/min/mL
Catalase	Eliminates hydrogen peroxide	1.84–26.1 nmol/min/mL
Glutathione peroxidase	Eliminates hydrogen peroxide	6.6–17.7 mM/min/L
Glutathione	Regeneration of other antioxidants	10.4–43.1 nmol/mg of protein
Melatonin	Free radical scavenger, antioxidant expression	<10–23 ng/L

Adapted from [18,26,41,54,55,56,57,60,61,62,63,64,66].

**Table 3 nutrients-11-01307-t003:** Growth factors in breastmilk.

Growth factors	Main Tissue Synthesized	Range (μg/mL)	Main Neonatal Functions
Epidermal-GF	Submandibular salivary gland	24–37	Intestinal mucosa maturation and healing, nutrient absorption, protein synthesis
Neuronal-GF	Cerebral cortex and hippocampus	2.8–934	Nervous system maturation, learning, and memory
Insulin-like-GF	Placenta and digestive system	5–35	Retinal vascularization, brain maturation
Vascular Endothelial-GF		505–650	Angiogenesis

Adapted from [20,71,80,85,86,87,90,99]; GF, growth factors

**Table 4 nutrients-11-01307-t004:** Adipokines in human breastmilk.

Adipokines	Tissue Synthesized	Range (ng/mL)	Preterm Infants	Main Neonatal Functions
Leptin	White adipose Placenta Mammary	0.2–10.1	↑/*?*	Anorexigenic T-lymphocyte responses
Adiponectin	Adipocytes	4.2–87.9	≈*/?*	Orexigenic Regulation of lipid/glucose metabolism Improvement of insulin sensitivity Anti-inflammatory actions
Resistin	Immune cells Epithelial cells	0.2–1.8	↑*/?*	Regulation of glucose homeostasis Inhibition of adipocyte differentiation Inflammatory response
Ghrelin	Stomach Pituitary Other	0.07–6	*?*	Orexigenic Gastric motility and secretion Adipogenesis Anti-inflammatory actions
Obestatin	Stomach Small intestine	0.4–1.3	*?*	Anorexigenic Body weight regulation
Nesfatin	Neurons Pancreas Other	0.008–0.01	*?*	Anorexigenic Production of body fat
Apelin	Heart Lung Other	43–81	*?*	Regulation of cardiovascular system Fluid homeostasis Angiogenesis Regulation of insulin secretion

Adapted from Catli et al. (2014) [102]. Arrows indicate the comparison with term infants (↑, higher; ↓, lower; ≈, similar); ? indicates unavailable or inconclusive data and the need for research.

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
