# Peer review of "A Review of Bioactive Factors in Human Breastmilk: A Focus on Prematurity"

_nutrients, 2019, doi:10.3390/nu11061307_

Reviewer 1 Report

I read with interest the manuscript of A Gila-Diaz & al reviewing the bioactive factors in human breast milk and suggest that the concentration of a few bioactive factors could be higher in preterm than in term own mother milk (OMM).

It is an extensive and well documented review on the bioactive components of human milk and their potential beneficial roles in preterm and term infants. In addition, the authors tended to evaluate the potential difference of those bioactive factors between preterm and term own mother milk and their potential benefit for preterm infants.

Unfortunately, these differences are still controversial and their potential benefit still speculative not supported by clinical evidence.

  Preterm human milk is the result of a premature stimulation of an immature breast, and the difference in composition could be the result of this unphysiological stimulation (premature stimulation) and not really dedicate to improve the postnatal adaptation of VLBW infants. Indeed, the nutritional composition is not specially adapted to preterm infants and the limited difference observed with term human milk disappears rapidly during the early weeks of lactation.

We regret that before the conclusion, there is not a clear summary of the main difference and the potential or clinical benefits that could be expected for the VLBW infants.

In the present review, some effects of the pasteurisation have been suggested but there is no clear clinical evidence on short- and long-term outcomes that feeding raw OMM versus pasteurized OMM provided any significant clinical benefit in VLBW infants except for a potential limited benefit on weight gain.

Similarly, a summary of the potential effect of pasteurization on the bioactive factors of preterm and term OMM could improve the interest of this review.

Feeding raw OMM in Very and Extremely LBW infants had some limitations as it has been shown that raw OMM provided to NICU could be heavy contaminated by pathogenic bacteria that could be deleterious for immature VLBW infants, inducing severe septicaemia and dead. Similarly, CMV reactivation in breastfeeding mothers and CMV transmission to V&ELBW infants is well documented with respiratory distress, NEC, septicaemia, dead or neurodevelopmental delay.

Therefore, the use of OMM needs to be promote in VLBW infants but an individual benefit/ risk ratio needs also to be estimated to decide its use as raw or pasteurized OMM.

Author Response

We would to thank for the useful comments. We have extensively revised the manuscript and we have included your suggestions.  

Reviewer 2 Report

The dilemma with prematurity is many more babies are surviving preterm birth, and especially those delivered before 28 weeks.  There is a need for nutrition support that allows for normal growth and development.  An emerging area of interest is focused on the multitudes of bioactive molecules present in breast milk that are considered to play multiple roles in mediating growth, development, and disease resistance.  It is more than meeting energy and macronutrient needs.  Understanding the normal changes in the types and concentrations of bioactive molecules in breast milk after preterm and term delivery will aid in better addressing the need of preterm and term infants.  From this perspective, the present contribution addresses a critical need.  The following comments are provided for the authors to consider how they might further improve their contribution.

Minor comment:  The manuscript has numerous grammatical mistakes, which will need correcting.

Major comments:

1.       How does this review add to what is already published? 

2.       A review of the published literature using PubMed and the terms “bioactive, breast milk, preterm” revealed a number of important publications that were not included in the review.  This was disconcerting for this reviewer.

3.       Along with that, the authors considered only some of the molecules in breast milk that are considered to be bioactive.  As examples:

a.       Phospholipids and associated fatty acids

b.      Proteins and peptides; lactoferrin, proteases, etc.

c.       Human milk oligosaccharides

d.      Milk fat globules

e.      Other hormones, such as cortisol

f.        Antibodies and other immune-associated immune factors

g.       microRNAs

h.      exosomes

The review should either be extensive or restricted to just a few, with that evident in the title. 

4.       There is too much background information in the introduction that is so very well-known and can be cited.  Some information is repeated (e.g., lines 51-52) The introduction could be reduced to<50% of the present length and focused on the main topic---changes in the bioactive molecules known to be in breast milk.

5.       An important consideration is that mature BM is not adequate for preterm infants.  This is why BM is supplemented using fortifiers.  However, those do not include bioactive molecules. 

6.       Much of the background information on antioxidants can be summarized and readers referred to specific publications.  As written the lengthy description for each of the antioxidants detracts from the main message---how they change during lactation.

7.       A table for each major section of bioactive type would be useful.  It could include the specific bioactive, references for functions, and how it changes from preterm to term lactation.

8.       What about colostrum?  How might it fit into the continuum?  Is preterm BM just immature colostrum?

9.       Lines 218-219. The cause of postnatal growth retardation is likely to be much more the result of inadequate nutrition support rather than prematurity itself.  Many preterm infants do not achieve adequate nutrition for days after delivery, causing a growth check and no doubt impacting development.   

10.   Table 1 is valuable in showing how preterm and term milk differ, and how formulas are supplemented, often to what might be considered excessive levels.  That is worth mentioning in greater detail.

11.   The controversy regarding the antioxidant content of preterm and term milk could be summarized in a table, allowing readers (and reviewers) to rapidly assess and recognize the differences.  The text could expand on the discrepancies.

12.   Several of the antioxidants are described, but without comparisons of preterm vs term milk.  The important point is not so much what they do, but that there is apparently not enough known about changes during lactation.  If this is true, the review can emphasize the need for more research. 

13.   Same for the growth factors—provide a short summary for each and refer readers to publications for more information, and dedicate effort to the differences during lactation. 

14.   Table 2 could be adapted for each section of bioactive molecule type, with the inclusion of a scoring system to indicate if preterm milk is higher, less, or the same as term. 

15.   There is so much more to do and know.  The first step in the process is to fully understand what bioactives are present in BM at different stages of lactation.  This review starts that step, but needs to be more comprehensive and complete.  The foundation is there and the authors are encouraged to expand and produce a contribution that will truly add to the knowledge as well as encourage the research that is needed. 

16. Might the authors suggest needs for future research? 

Author Response

We would to thank for usuful comments. We have extensively revied and we have included your suggestions.
